# A Machine Learning Model to Predict Length of Stay and Mortality among Diabetes and Hypertension Inpatients

**DOI:** 10.3390/medicina58111568

**Published:** 2022-10-31

**Authors:** Diana Barsasella, Karamo Bah, Pratik Mishra, Mohy Uddin, Eshita Dhar, Dewi Lena Suryani, Dedi Setiadi, Imas Masturoh, Ida Sugiarti, Jitendra Jonnagaddala, Shabbir Syed-Abdul

**Affiliations:** 1Graduate Institute of Biomedical Informatics, College of Medical Science and Technology, Taipei Medical University, Taipei 106, Taiwan; 2International Center for Health Information Technology (ICHIT), College of Medical Science and Technology, Taipei Medical University, Taipei 106, Taiwan; 3Department of Medical Record and Health Information, Health Polytechnic of the Ministry of Health Tasikmalaya, Tasikmalaya 46115, West Java, Indonesia; 4CGD Health Pty Ltd., Throsby, ACT 2914, Australia; 5Research Quality Management Section, King Abdullah International Medical Research Center, Ministry of National Guard-Health Affairs, Riyadh 11481, Saudi Arabia; 6School of Population Health, University of New South Wales, Kensington, NSW 2033, Australia; 7School of Gerontology and Long-Term Care, College of Nursing, Taipei Medical University, Taipei 106, Taiwan

**Keywords:** predictive modeling, external validation, length of stay, mortality, type 2 diabetes, hypertension, machine learning

## Abstract

*Background and Objectives*: Taiwan is among the nations with the highest rates of Type 2 Diabetes Mellitus (T2DM) and Hypertension (HTN). As more cases are reported each year, there is a rise in hospital admissions for people seeking medical attention. This creates a burden on hospitals and affects the overall management and administration of the hospitals. Hence, this study aimed to develop a machine learning (ML) model to predict the Length of Stay (LoS) and mortality among T2DM and HTN inpatients. *Materials and Methods*: Using Taiwan’s National Health Insurance Research Database (NHIRD), this cohort study consisted of 58,618 patients, where 25,868 had T2DM, 32,750 had HTN, and 6419 had both T2DM and HTN. We analyzed the data with different machine learning models for the prediction of LoS and mortality. The evaluation was done by plotting descriptive statistical graphs, feature importance, precision-recall curve, accuracy plots, and AUC. The training and testing data were set at a ratio of 8:2 before applying ML algorithms. *Results*: XGBoost showed the best performance in predicting LoS (R2 0.633; RMSE 0.386; MAE 0.123), and RF resulted in a slightly lower performance (R2 0.591; RMSE 0.401; MAE 0.027). Logistic Regression (LoR) performed the best in predicting mortality (CV Score 0.9779; Test Score 0.9728; Precision 0.9432; Recall 0.9786; AUC 0.97 and AUPR 0.93), closely followed by Ridge Classifier (CV Score 0.9736; Test Score 0.9692; Precision 0.9312; Recall 0.9463; AUC 0.94 and AUPR 0.89). *Conclusions*: We developed a robust prediction model for LoS and mortality of T2DM and HTN inpatients. Linear Regression showed the best performance for LoS, and Logistic Regression performed the best in predicting mortality. The results showed that ML algorithms can not only help healthcare professionals in data-driven decision-making but can also facilitate early intervention and resource planning.

## 1. Introduction

Non-Communicable Diseases (NCDs), such as Type 2 Diabetes Mellitus (T2DM) and Hypertension (HTN), are a major public health problem and a leading cause of mortality worldwide. They pose great economic threats and burdens due to their treatment cost and complications [1]. Mainly triggered by obesity, fatty food, physical inactivity, and a sedentary lifestyle, T2DM is one of the most common NCDs [2]. The number of cases and prevalence of diabetes have continued to increase over the last few decades, as approximately 422 million people have diabetes worldwide, with the majority living in low and middle-income countries [3]. Each year, diabetes is directly responsible for 1.6 million deaths, and an estimated 193 million people are diabetic but are unaware of it [4]. T2DM and obesity are the leading factors for the global prevalence of Hypertension (HTN). HTN is one of the silent killer NCDs because sometimes, people with HTN do not manifest signs and symptoms [5]. It is a major risk factor for cardiovascular, brain, kidney, and other diseases. The prevalence of HTN increases with age, as an estimated 1.2 billion adults aged 30–79 have HTN worldwide, with a significant ratio in low and middle-income countries, and approximately 46% of people are unaware of having this condition [6].

In the literature, various studies have been conducted to assess the treatment/outcomes of T2DM and HTN patients. Hospitalized patients, mostly with hypertensive emergencies or urgency, who then sporadically exhibit acute HTN and who are deemed worthy of clinical attention, may also have chronic HTN [7]. Cases of patients with HTN are common among diabetes, with prevalence depending on the type and duration of diabetes, age, sex, race/ethnicity, BMI, history of glycemic control, and presence of kidney disease, among other factors [8]. HTN is also a major cause of morbidity and mortality for individuals with diabetes. More than 50% of patients with HTN also have DM [9]. T2DM increases the risks of heart failure and mortality in patients with HTN. Given their common risk factors, HTN and T2DM often coexist. In general, HTN is prevalent among 70% of T2DM patients, whereas patients with HTN are 2.5 times more likely to develop T2DM as a primary comorbidity [10,11,12].

The Length of Stay (LoS) is the amount of time a patient stays in the hospital after being admitted due to a medical condition and is regarded as one of the most important metrics for hospital administration and management [13]. Several studies have shown that LoS is associated with other clinical outcomes; for example, if the patient remains in the ICU for more than three days, he is more likely to die [14]. However, Lingsma et al. [15] indicated that there is a direct correlation between LoS and mortality during the index admission, and Sud et al. [16] showed that prolonged LoS is associated with higher rates of mortality and readmission.

Taiwan implemented the National Health Insurance (NHI) system in 1995, and it has a high coverage and utilization rate. However, the healthcare system in Taiwan is facing immense challenges due to rapid population aging [17]. Approximately 9996 people in Taiwan died from DM in 2019, with 2736 of the deaths being recorded to be among people 85 years of age and above [18]. The number of deaths due to HTN in 2019 was 6255, and the majority of the cases had an age of 85 years or more [18]. Taiwan is among the nations with the highest rates of Type 2 Diabetes Mellitus (T2DM) and Hypertension (HTN). As more cases are reported each year, there is a rise in hospital admissions for people seeking medical attention. This creates a burden on the hospitals and affects the overall management and administration of the hospitals. Accurate identification of patients enables early planning of treatment and provision of more intensive care to accelerate their recovery, intervention, and improvement of clinical outcomes, thereby reducing LoS as well as improving the planning and resource management [19].

Artificial Intelligence (AI) is an innovative field of computer science that has transformed the practice of medicine and reshaped the delivery of healthcare. With the latest surge of AI in healthcare, one of its powerful domains, Machine Learning (ML), has extensively been used for improving the accuracy, prediction, and quality of work in this domain [20]. An important application of ML algorithms used in hospitals is the precise prediction of mortality and LoS, which in turn can classify patients with different risk factors of outcome [21]. Accurate LoS prediction of inpatients is not only important in improving patient care but is also critical for resource management and planning in hospitals [13]. Therefore, the objective of this study was to utilize ML algorithms in order to predict the LoS and mortality of patients diagnosed with T2DM and HTN using the data from Taiwan’s National Health Insurance Research Database (NHIRD). The National Health Insurance Research Database (NHIRD) of Taiwan is a unique and large national database that has been widely used as an excellent resource for scientific research in healthcare, along with its other benefits and purposes [22,23].

## 2. Materials and Methods

### 2.1. Data Source

Taiwan began its healthcare reforms in the 1980s, following two decades of rapid economic growth. The National Health Insurance (NHI) Act was passed on 19 July 1994, and the NHI model was adopted. The Bureau of NHI developed NHIRD to support data-driven decision and policy making [24]. NHIRD is a cohort of registry and claims data of all 23 million residents of Taiwan [19]. NHIRD data were made available to researchers for the period of 2000–2013. We have taken the four years’ latest releases, i.e., data from 2010–2013, in our study. We queried NHIRD for participant user files to conduct this retrospective cohort study. Data included sex, age, birthdates, discharge status, treatment, status change indicator, death, hospital cost, LoS, etc.

Our study population consisted of 65,037 patients. However, there were patients with T2DM (*n* = 25,868), HTN (*n* = 32,750), and both T2DM and HTN (*n* = 6419). In general, 70% of patients with T2DM had HTN, and patients with previous HTN were 2.5 times more likely to develop T2DM [3].

### 2.2. Inclusion and Exclusion Criteria

We included patients with T2DM or HTN.

The inclusion criteria were as follows. (i) Patients diagnosed with T2DM using the following International Classification of Diseases-9 (ICD-9) Revision codes: (25000, 25002, 25010, 25012, 25020, 25022, 25030, 25032, 25040, 25042, 25050, 25052, 25060, 25062, 25070, 25072, 25080, 25082, 25090, 25092) or (ii) Patients diagnosed with HTN using the following ICD-9 Revision codes: (3482, 36504, 4010, 4011, 4019, 40501, 40509, 40511, 40519, 40591, 40599, 4160, 45930, 45931, 45932, 45933, 45939, 5723, 64200, 64201, 64202, 64203, 64204, 64210, 64211, 64212, 64213, 64214, 64220, 64221, 64222, 64223, 64224, 64230, 64231, 64232, 64233, 64234, 64270, 64271, 64272, 64273, 64274, 64290, 64291, 64292, 64293, 64294).

Patients with both T2DM and HTN were identified by querying the above two datasets. Finally, we selected patients with either T2DM or HTN as primary comorbidities for predicting LoS and mortality. We also excluded patients with duplicate records, patients who died on discharge, those with missing/incomplete data, patients who died on the day of admission, and deaths due to injuries or suicide. Demographic characteristics of all patients with T2DM, HTN, and both HTN and T2DM has been explained in the results. However, we continued to develop the prediction model by previously excluding patients who had both T2DM and HTN (see Figure 1).

### 2.3. Predictors and Outcomes

The outcome of interest in our study included mortality and LoS. The outcome predicted using the model for mortality was a categorical variable with the values 1 = “alive” or 0 = “death”, and LoS was predicted as a continuous variable. All the selected predictors were based on data obtained before discharge. A total of 67 predictor variables consisting of hospital cost, vital signs and symptoms, comorbidities, and demographic characteristics were extracted from NHIRD.

The covariates of interest in our study included Gender, Age, Discharge status, HTN, T2DM, Number of comorbidities, Hospital cost, LoS, Days spent in acute bed, Days spent in chronic bed, Transfer code, Case classification, Pneumonia, Urinary tract infection (UTI), Cellulitis, Congestive heart failure, Inguinal hernia, Acute pancreatitis, Aneurysm, Hearing, LoS, Hypertrophy, Acute pyelonephritis, Cerebral artery hemorrhage, Intracerebral hemorrhage, Congestive heart failure, Calculus of urethra, Obstructive chronic bronchitis, Displacement of the lumbar vertebral disc, and Malignant neoplasm of liver. All were analyzed using EDA plots and descriptive statistics.

### 2.4. Handling Missing Values

As with most clinical data, NHIRD data contained a significant number of missing values. Initially, all the variables that were not selected for inclusion in the study were removed. Thereafter, we examined the proportion of missing values in each of the candidate variables. The overall missingness in each of the features was less than 10%, so we ultimately removed all the missing values from our study.

### 2.5. Features Selection

In our study, we selected predictors based on literature review, expert opinion, and univariate and bivariate analysis [25]. First, we identified features through expert opinions and a literature review. Thereafter, we conducted univariate and bivariate analyses on the feature set using chi-square. Subsequently, 24 features were selected for the LoS prediction, and 27 features were considered for the mortality prediction (see Figure 2).

### 2.6. Managing Class Imbalance

Accuracy is one of the most commonly used metrics to evaluate ML models. This measure is usually not sufficient when the data are highly imbalanced (as in the case of our study, the variance between survivors and the mortality was considerably high). However, the nature of our prediction problem required a high rate of correct detection of the mortality of patients. The most commonly used methods in many types of research to solve the class imbalance problem are oversampling the minority class [26], under-sampling the majority class [27], or a combination of both [24]. However, under-sampling may cause the loss of vital information by removing significant patterns, and similarly, over-sampling may cause overfitting and introduce additional computational tasks. To solve this problem, Chawla et al. [28] introduced a Synthetic Minority Over-sampling Technique (SMOTE) by generating a synthetic example rather than replacement with replication. Our study used a combination of oversampling by SMOTE and under-sampling by Random Under Sampler to address the class imbalance, and this combination gave us very good results in predicting mortality.

### 2.7. Predictive Model Development and Evaluation

Once the data were preprocessed, it was split into train and test datasets, and prediction algorithms were applied. We tested and evaluated various ML algorithms before fine-tuning the model hyperparameters. For the classification problem, we tested with a Decision Tree Classifier, Random Forest Classifier, Logistic Regression, AdaBoost Classifier, Bagging Classifier, Gradient Boosting Classifier, XGBoost Classifier, Support Vector Machines, K-Neighbors Classifier, and Naïve Bayes. After evaluation, we shortlisted a set of algorithms for hyperparameter tuning, namely Logistic Regression (LoR), Ridge Classifier (RC), Gradient Boosting Classifier (GBC), Bagging Classifier (BC), K-Neighbors Classifier (KNN), Random Forest Classifier (RFC), and Support Vector Machine (SVM) to predict mortality. For the regression problem, we used Linear Regression (LR), Support Vector Machine (SVM), Extreme Gradient Boosting (XGBoost), Gradient Boosting Machine (GBM), and Random Forest (RF) to predict LoS based on the patient characteristics described in predictors and outcome section. Other than predicting the response from a set of predictors, another common important step in data-driven modeling is to identify which predictors are most relevant to the prediction task and the contribution of each feature in predicting the outcome. A grid search was set up for each combination of hyperparameters, and the best combination was selected by comparing scores from nested and non-nested 10-fold cross-validation procedures. After the selection of optimal tuning parameters, these were then used to train and evaluate the algorithms through nested 10-fold cross-validation. Ten percent of the training portion of each cross-validation was set aside for selecting the optimal classification threshold and the rest for final evaluation. The Receiver Operating Characteristics (ROC), Area Under the Curve (AUC), F-beta, Precision, Recall, Cross Validation score, Accuracy score, Balanced Accuracy score, Test score, and Area Under Precision-Recall (AUPR) were used to evaluate the models. Both mortality and LoS datasets were partitioned into training and testing sets in an 8:2 ratio. The training set was used to run the model, and the testing set was used to determine the performance of the model after learning.

### 2.8. Length of Stay (LoS)

We used several models to predict LoS, including SVM, LR, GBM, XGBoost, and RF. The parameters chosen using the grid search for each of the algorithms were GBM (cost: 0, mstop: 37, NU 0.68, num_sample: 34), RF model (ntree: 210, mtry: 8, node size: 49), LR model (alpha: 0.19, S: 11.6, and n lambda: 19), XGBoost model (eta: 0.04, gamma: 3.9, max depth: 10, and subsample: 0.65), and SVM model (C: 9.2). The rest of the parameters in each model were set to default.

### 2.9. Mortality

The models for predicting mortality included LoR, RC, SVM, RFC, KNN, BC, and GBC. The parameters chosen using the grid search for each of the algorithms were LoR (C: 1000, solver: newton-cg), RC (alpha: 0.1), RFC (max_features: sqrt, n_estimators: 100), KNN (metric: manhattan, weights: distance), BC (n_estimators: 10), and GBC (learning_rate: 0.001, max_depth: 3, n_estimators: 10, subsample: 0.5). The rest of the parameters in each model were set to default.

### 2.10. Document Software and Libraries

We used string R version 4.2.1 and Python 3.7 for the development of the LoS prediction model. The mortality prediction model and cross-validation analysis were performed by using Python v3.8.8, Anaconda v1.7.2, Jupyter core v4.7.1, Jupyter-notebook v6.3.0, Qt console v5.0.3, Ipython v7.22.0, and ipykernel v5.3.4. The libraries used in the R package were MASS, Tidyverse, Mlr, XGBoost, kernlab and random forest. The libraries used in Python were SKLearn, CRAN, LoR Library, TensorFlow, XGBoost, Numpy, pandas, Matplotlib, seaborn, imblearn, and collections.

## 3. Results

### 3.1. Patient Characteristics

A total of 65,037 patients were descriptively analyzed, as shown in Table 1. For the predictive analysis, we excluded the patients having both T2DM and HTN and considered the cohort with a total of 58,618 patients that included the patients with either T2DM and HTN only. The mean age of the cohort was 75.12 ± 13.65 years. Over half of the included patients were male (51.16%), and half of them were between the age of 58 and 80 (50.26%). Patients with only HTN (55.87%) had a higher proportion than patients with only T2DM (44.13%). The average LoS for patients with only T2DM (8.46 days) was higher than those for patients with only HTN (6.56 days). Patients with T2DM had a higher mortality rate (2.26%) than those with HTN (0.91%). The demographic characteristics of overall patients are shown in Table 1.

### 3.2. Features Selection

The following 24 variables were selected for LoS prediction: Gender, Closed fracture of unspecified part of neck of femur, Age, Age categorical, Pneumonia, Diabetes, Hypertension, Cerebral artery occlusion, UTI, Cellulitis, Intracerebral hemorrhage, Congestive heart failure, Hearing loss, Acute pyelonephritis, Acute pancreatitis, Aneurysm, Osteoarthrosis, Calculus of ureter, Inguinal hernia, Obstructive chronic bronchitis, Hypertrophy, Malignant neoplasm of the liver, and Displacement of a lumbar intervertebral disc.

The following 27 variables were selected for mortality prediction: Days in acute bed, Days in chronic bed, Transfer code, Case classification, Gender, LOS, Age, Age group, Pneumonia, Diabetes, Hypertension, Cerebral artery occlusion, UTI, Cellulitis, Intracerebral hemorrhage, Congestive heart failure, Hearing loss, Acute pyelonephritis, Acute pancreatitis, Aneurysm, Osteoarthrosis, Calculus of ureter, Inguinal hernia, Obstructive chronic bronchitis, Hypertrophy, Malignant neoplasm of the liver, Displacement of a lumbar intervertebral disc, Closed fracture of unspecified part of neck or femur.

### 3.3. Comorbidities of T2DM and HTN

The most common comorbidities among patients with T2DM and HTN from our data are shown in Table 2. Metabolic disorders were the highest of the most common comorbidities of inpatients with T2DM and HTN.

### 3.4. Length of Stay (LoS)

We evaluated various metrics for LoS prediction. The performance of all the models is presented in Table 3. The best-performing model was XGBoost with R2 of 0.633, followed by RMSE 0.386, MAE 0.123, and MSE 0.312.

### 3.5. Feature Importance

A list of the top 15 features’ importance plot in LoS prediction is shown in Figure 3. The figure shows that age is the most important feature in LoS prediction using chi-square. The other most important features influencing the LoS prediction were gender and diabetes as co-morbidity.

### 3.6. Mortality

Table 4 indicates the mortality prediction performance. The best results were determined by using RF with an AUROC of 0.996.

### 3.7. Feature Importance

The top 15 features’ importance scores in mortality prediction are shown in Figure 4. We selected all the features to predict mortality. The figure shows that the displacement of a lumbar intervertebral disc was the most important feature in mortality prediction using Random Forest Classifier. This comorbidity could be the major factor affecting mortality. The other most important features influencing mortality prediction were cerebral artery occlusion and age.

### 3.8. The Accuracy and LoS Plots

Figure 5 and Figure 6 show the accuracy and LoS plot for mortality prediction by using a neural network model. We developed a neural network to produce the accuracy and LoS plots on a training dataset. We obtained both accuracy and LoS plot values to evaluate the performance of the classification of subjects on each iteration. However, accuracy is a parameter that may present any bias in the data. Therefore, to confirm that the classification of subjects was statistically significant, we also estimate other parameter metrics. The LoS plot in Figure 5 shows that the model is relatively good since the dataset is unbalanced; however, the model needs to learn more.

### 3.9. The AUC and Precision-Recall Curves

The Figure 7, Figure 8, Figure 9, Figure 10, Figure 11 and Figure 12 represent the AUC and Precision-recall curves for the different models used for prediction of LoS and mortality.

### 3.10. Calibration

The Figure 13, Figure 14, Figure 15 and Figure 16 shows the calibration of different models used for prediction of LoS and mortality.

## 4. Cross–Validation

The results from nested and non-nested cross-validation on the training dataset are compared in the figures below (see Figure 17, Figure 18, Figure 19, Figure 20 and Figure 21). We conducted five trials and compared the nested and non-nested cross-validation scores as well as the average difference in scores from each experiment. The *x*-axis and *y*-axis represented the individual trial # (# depicts number) and score, respectively. This was then applied to all the algorithms. We observed that the average difference between trials was noticeably small, i.e., the difference between nested and non-nested cross-validation scores was not much. 

## 5. Discussion

Earlier, we conducted a similar study using several machine learning techniques to predict LoS and mortality for patients diagnosed with T2DM and HTN in Indonesia [29]. Our previous study had the same objectives as the current one, but it was conducted using an Indonesian insurance claim-based dataset called Indonesia Case-Based Groups (INA-CBGs) from a state-owned type B regional public hospital in Tasikmalaya, the Dr. Soekardjo Regional Public Hospital (RSUD Dr. Soekardjo), with a sample size of 4376 patients. Our current study was conducted using Taiwan’s NHIRD data using a greater sample size of 65,037 patients. The advantage of the current study is the NHIRD’s data, which are made up of multiple hospitals and healthcare service clinics, and it is the best representation of the national population as it covers more than 99% of the resident population of Taiwan. In comparison to our previous study results, where LR and GBM models best predicted LoS and MLP best predicted the mortality, the current study also showed that XGBoost had the best performance in predicting the patients’ LoS, along with RF, which had similar performance, while LoR performed the best in predicting mortality, closely followed by Ridge Classifier. The ML models in both of these two studies corroborate a good prediction of LoS and mortality among T2DM and HTN patients and hence, prove their utility in medical decision-making, patient safety, and hospital resource management.

In addition to our previous study, there is an abundance of other studies in the literature that utilize ML models for the prediction of diseases. For example, two of the studies used ML approaches for the prediction of LoS or mortality in diabetic patients [30,31], but neither of these studies predicted both LoS and mortality. Compared to that, in our study, LoS and mortality were predicted in order to enhance healthcare quality. The findings from our study revealed that the majority of the patients diagnosed with T2DM and HTN were male. Our findings differed from another study done in Taiwan that showed that women were more associated with HTN [30]. Our results showed that the majority of T2DM and HTN patients fall in the age group between 58 and 80, with the youngest patient being 35 years old. A population-based cross-sectional survey also found that the majority of the population aged 60 years and above were diagnosed with HTN in Taiwan [32]. Another study forecasted that the number of cases of diabetes in people aged ≥65 years will increase from 9.2 million in 2014 to 21 million in 2030 [33]. Although an increasing number of individuals with T1DM were old aged [34], this discussion of pathophysiology concerns T2DM, the most common incident and prevalent type in older age groups overwhelmingly, as older adults are at high risk for the development of T2DM due to the combined effects of increasing insulin resistance and impaired pancreatic islet function with aging.

Our study revealed that the discharge status of a large number of patients with HTN and T2DM was at the end of transfer in outpatient treatment. Comorbidity was also one of the factors affecting the outcome of a patient’s medical condition. Our findings revealed that the majority of patients with T2DM have at least three or more comorbidities, while patients with HTN have at least two comorbidities. The most common comorbidities in our study included metabolic disorders, coronary artery disease, myocardial infarction, stroke, and congestive heart failure. Another study also indicated that ischemic stroke is one of the major vascular complications of diabetes mellitus [35]. Atherosclerotic cardiovascular disease, including coronary heart disease, cerebrovascular disease, and peripheral arterial disease, is the major cause of death and disability in patients with T2DM [36]. Furthermore, T2DM is associated with an increased risk of multiple coexisting medical conditions in older adults, such as cardiovascular and microvascular diseases [37,38]. A group of conditions termed geriatric syndromes also occurs at higher frequency in older adults with T2DM and may affect self-care abilities and health outcomes, including quality of life [39].

Our current findings (see Table 1) indicated that the inpatient cost of patients with T2DM exceeds patients with HTN. The findings are consistent with a study by Mutsa P. Mutowo, who also showed that there was a higher median cost and interquartile range (IQR) for DM patients compared with HTN patients [40]. In a study done in Taiwan, the risk of hospitalization and healthcare cost associated with diabetes complication severity index in Taiwan’s NHIRD showed that inpatient costs constituted a large part of the total medical costs of DM and its complications [41]. In addition, it was found that the greater the number and severity level of T2DM complications, the higher risk of mortality and hospitalizations [42]. Furthermore, previous estimates of the costs associated with T2DM and its related problems in Taiwan have been based on (The Adapted Diabetes Complications Severity Index) DCSI scores rather than individual complications. The average inpatient LoS for T2DM patients was eight days; for HTN, it was approximately one week. Our results differ from a study done in Japan, where the mean LoS of DM patients ranged from 10.9 days to 15.1 days, depending on the patient’s age [43].

Evaluation metrics are an integral aspect of ML, as they are used as indicators to assess the performance of ML models. The most commonly used metrics are accuracy and error rate [24]; however, these metrics are not the best measures to use if you have data that are highly imbalanced, as the overall accuracy will be biased toward the majority class regardless of the minority class, which will consequently lead to poor performance.

From the literature, the majority of researchers have used oversampling since this method is capable of balancing class distributions without removing potentially critical majority examples [44]. One of the most common errors that most people make is applying oversampling to the entire original data, conducting cross-validation, and finally evaluating the model [45]. This error usually leads to building biased models and producing over-optimistic error estimates. One of the strengths of our study is that we performed a combination of oversampling (SMOTE) and under-sampling methods. This procedure was applied during nested and non-nested cross-validation, the dataset was first divided into *k* stratified partitions, and only the training set was oversampled. In this procedure, the observations included in the test set are never oversampled or seen by the model during the training stage, thus allowing a proper evaluation of the model’s capability to generalize. The top feature predictor is the displacement of a lumbar intervertebral disc in patients, which ultimately results in higher LoS and mortality. Several studies have been conducted in this area; for example, Sakellaridis et al. proved that patients operated on for lumbar disk disease have a statistically significant increased incidence of diabetes mellitus compared to similar patients operated on for other reasons [46].

## 6. Conclusions

In this study, we used ML algorithms to predict the LoS and mortality among T2DM and HTN patients. The results showed that the XGBoost was the best model for LoS, and LoR provided good results in mortality prediction. The low R2 score for LoS algorithms is a concern for practical use; we have taken LoS as a feature for mortality prediction and obtained good balanced accuracy. Therefore, we recommend that this model could be a possible prediction tool for medical decision-making. An accurate forecast of hospital stay and mortality enables early planning and treatment to improve patient’s clinical outcomes. It can also help with better resource allocation and availability of hospital beds. Our results lay the foundation for future work in developing rapid and robust classification and regression algorithms that can leverage the minimal amount of available data. Moreover, the combination of oversampling and under-sampling can be applied to the unbalanced dataset.

## Figures and Tables

**Figure 1 medicina-58-01568-f001:**
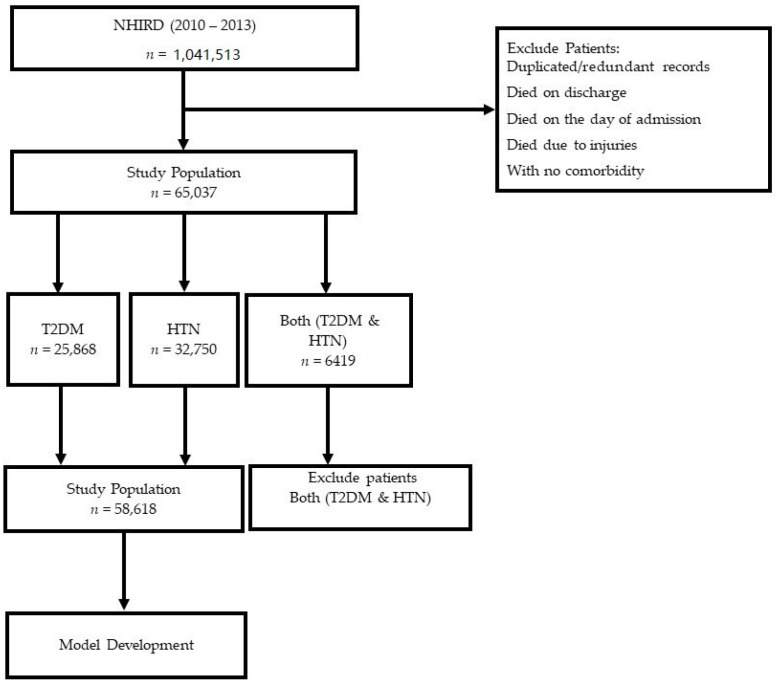
Flowchart illustrating patient selection process.

**Figure 2 medicina-58-01568-f002:**
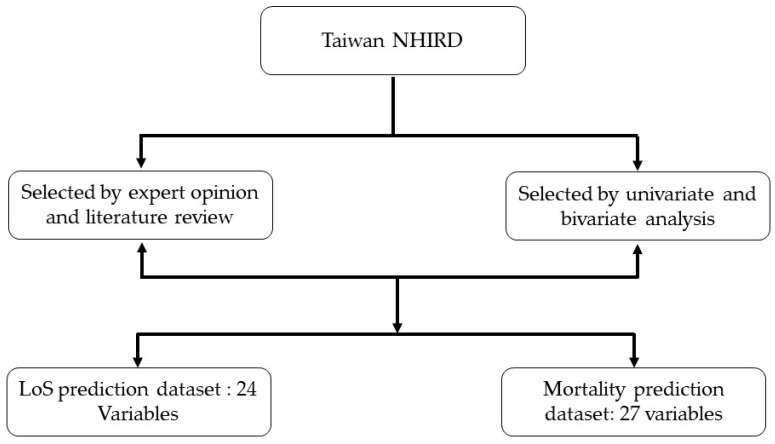
Feature selection process for LoS and mortality prediction.

**Figure 3 medicina-58-01568-f003:**
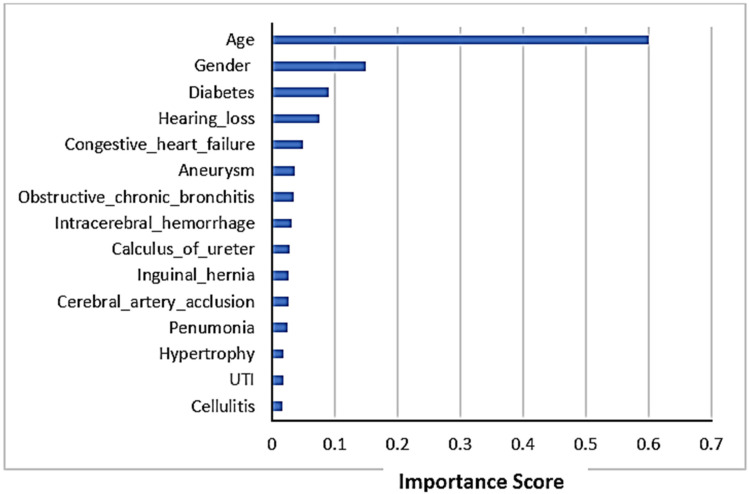
Feature importance scores in LoS prediction.

**Figure 4 medicina-58-01568-f004:**
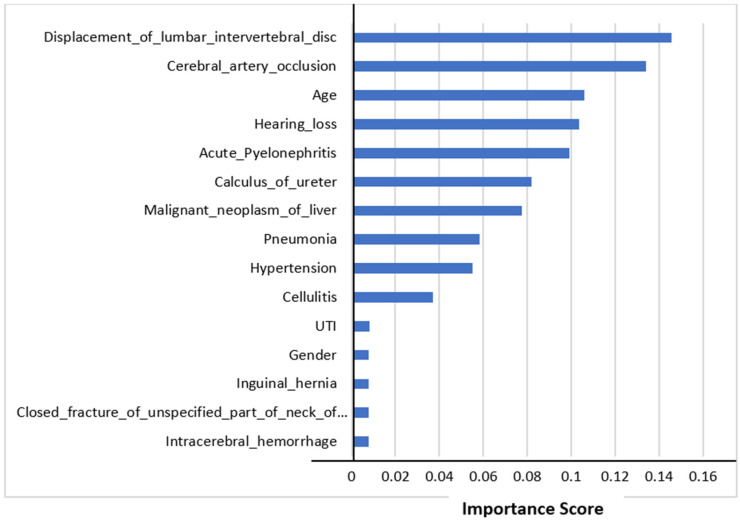
Feature importance scores in mortality prediction.

**Figure 5 medicina-58-01568-f005:**
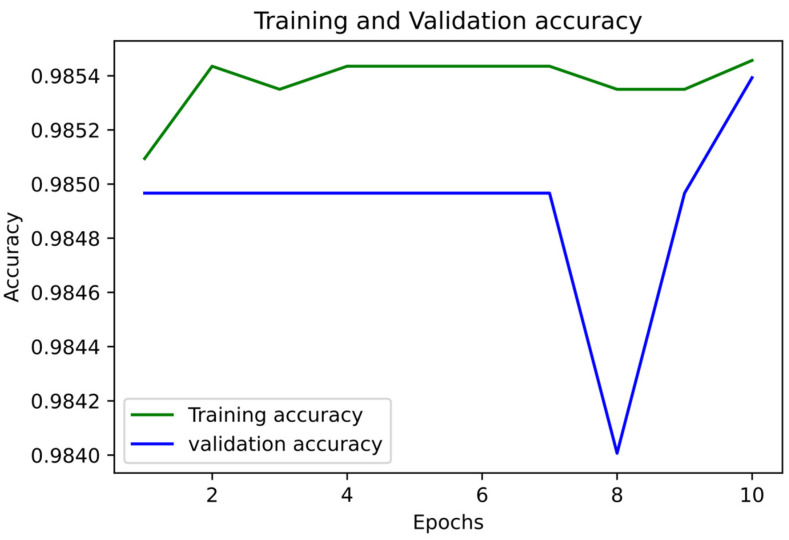
Accuracy Plot for Mortality Prediction.

**Figure 6 medicina-58-01568-f006:**
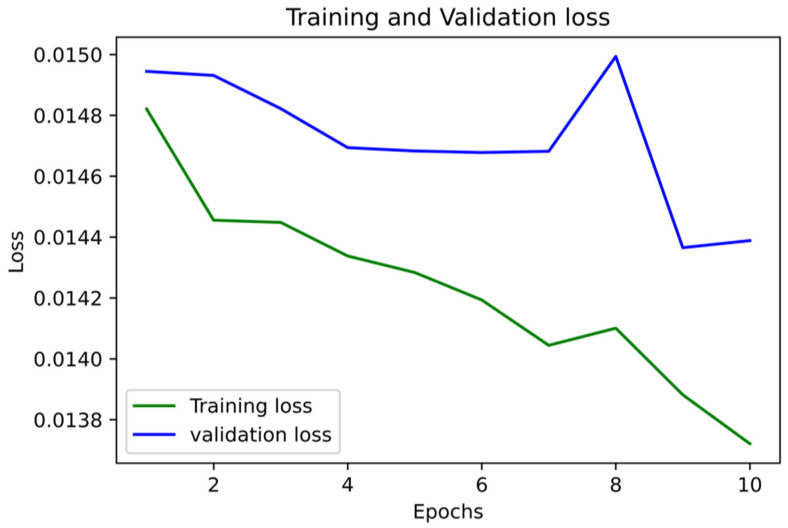
LoS Plot for Mortality Prediction.

**Figure 7 medicina-58-01568-f007:**
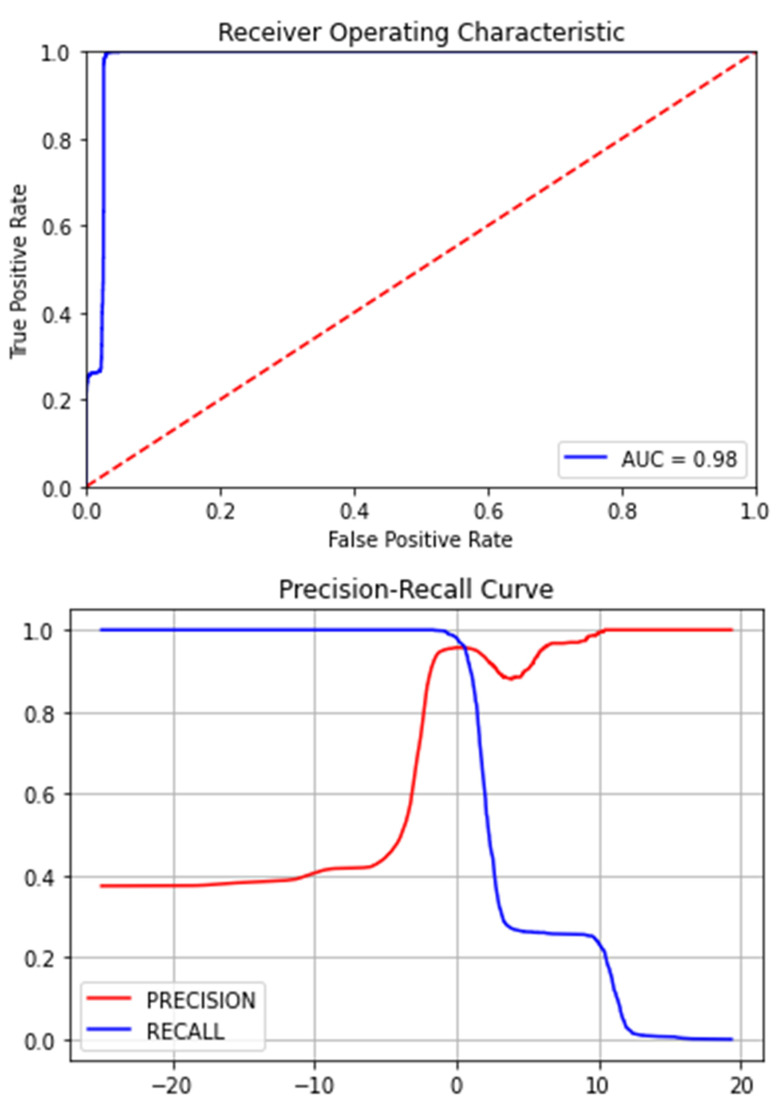
AUC and Precision-Recall Curve for Logistic Regression.

**Figure 8 medicina-58-01568-f008:**
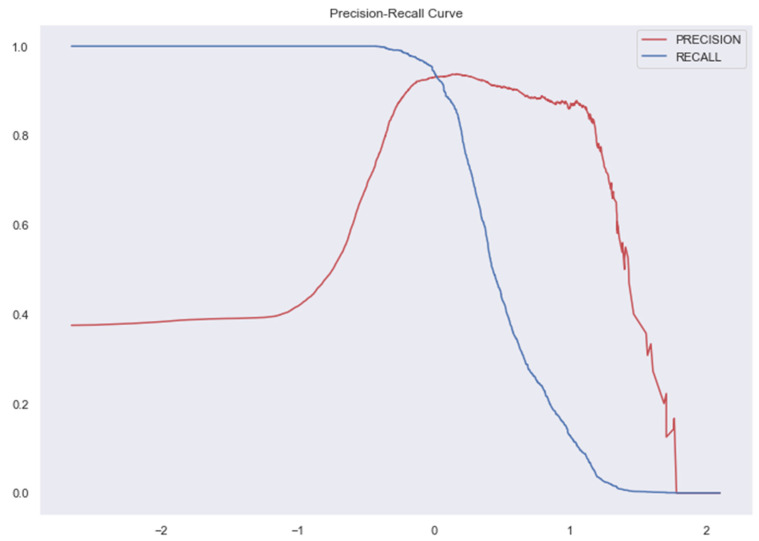
Precision-Recall Curve for Ridge Classifier. AUC for Ridge Classifier–0.94 (The AUC curve for RC cannot be plotted due to the nature of the algorithm.).

**Figure 9 medicina-58-01568-f009:**
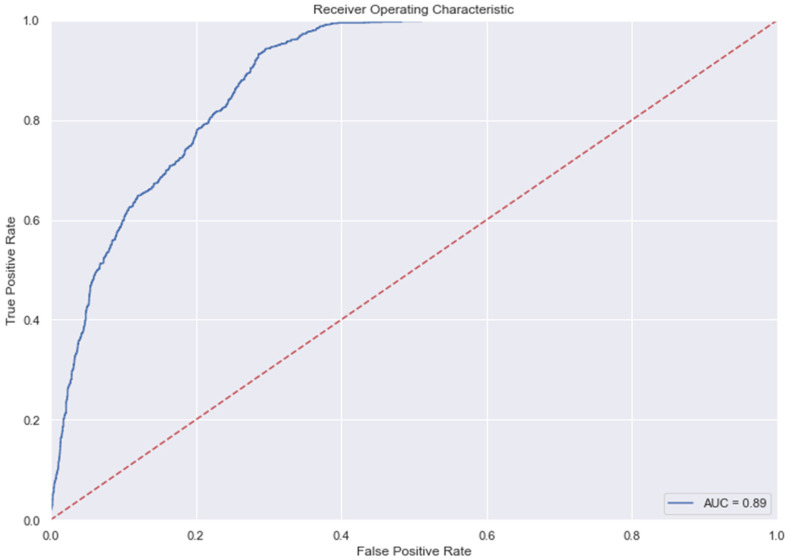
AUC and Precision-Recall Curve for SVM.

**Figure 10 medicina-58-01568-f010:**
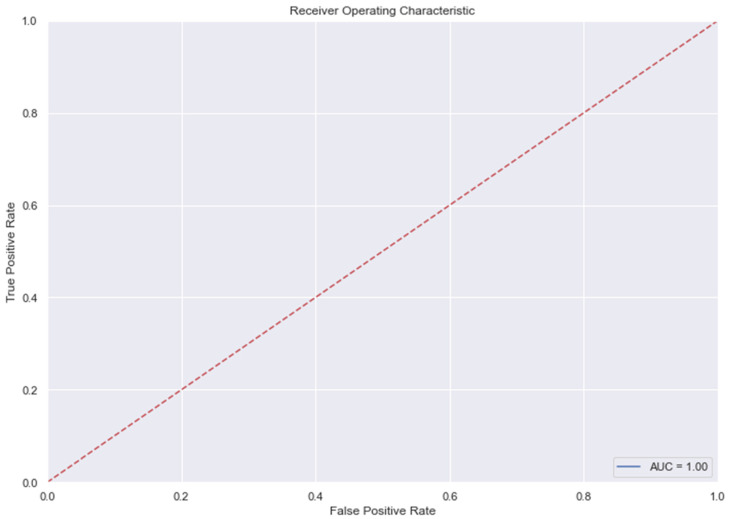
AUC and Precision-Recall Curve for Random Forest Classifier.

**Figure 11 medicina-58-01568-f011:**
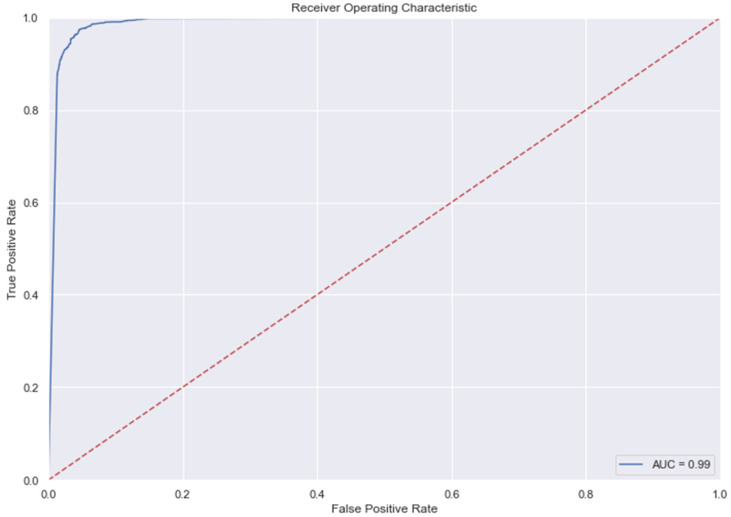
AUC and Precision-Recall Curve for KNN.

**Figure 12 medicina-58-01568-f012:**
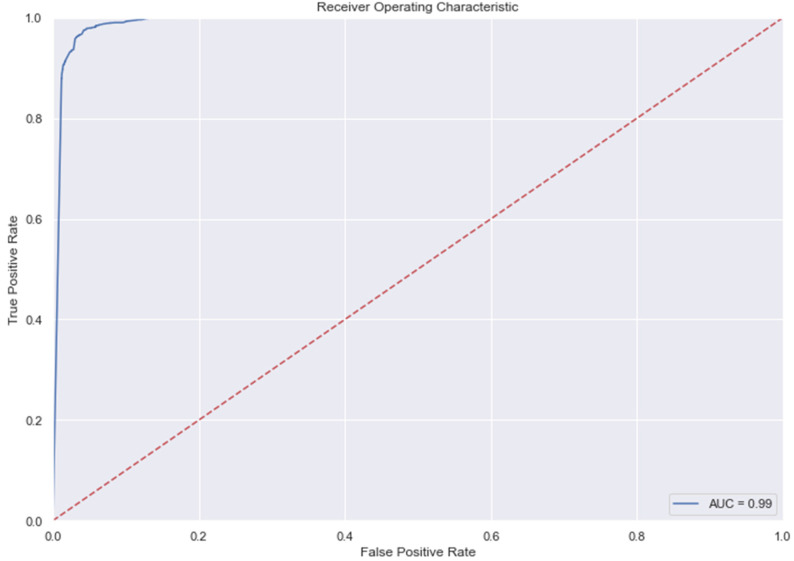
AUC and Precision-Recall Curve for Gradient Boosting Classifier.

**Figure 13 medicina-58-01568-f013:**
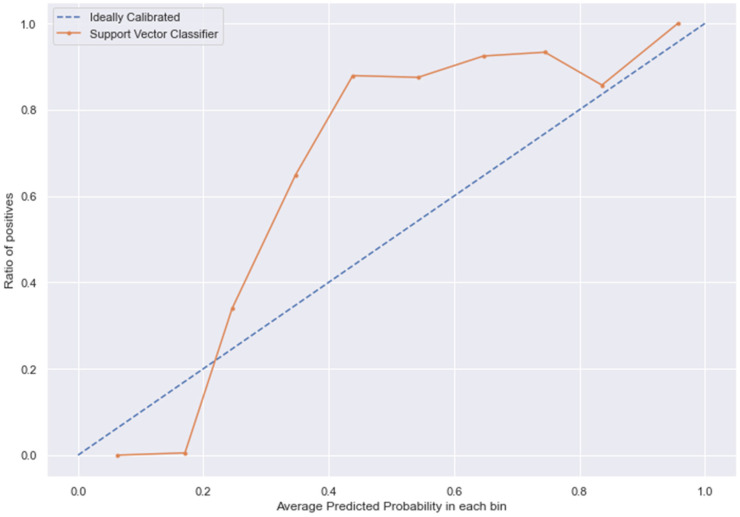
SVC line without calibration vs. after calibration (Ideal fit is represented as blue line).

**Figure 14 medicina-58-01568-f014:**
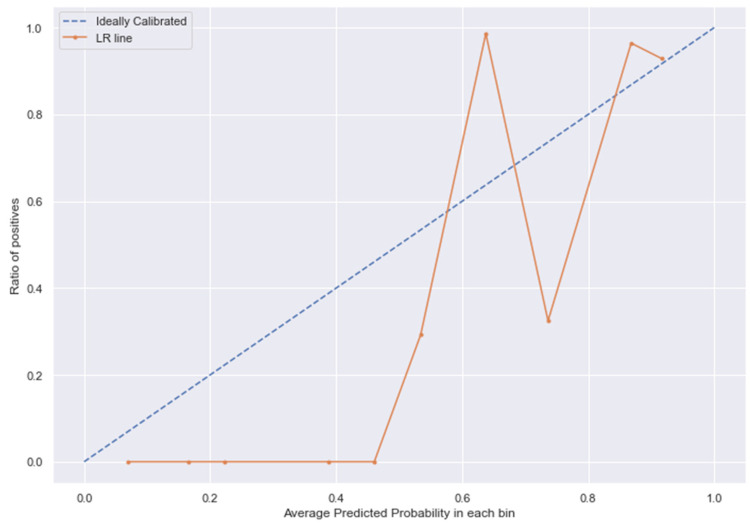
LR line without calibration vs. after calibration (Ideal fit is represented as blue line).

**Figure 15 medicina-58-01568-f015:**
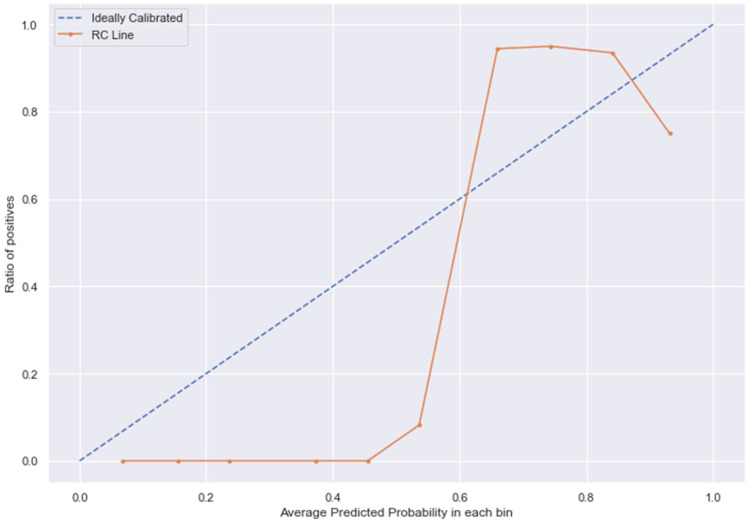
RC line without calibration vs. after calibration (Ideal fit is represented as blue line).

**Figure 16 medicina-58-01568-f016:**
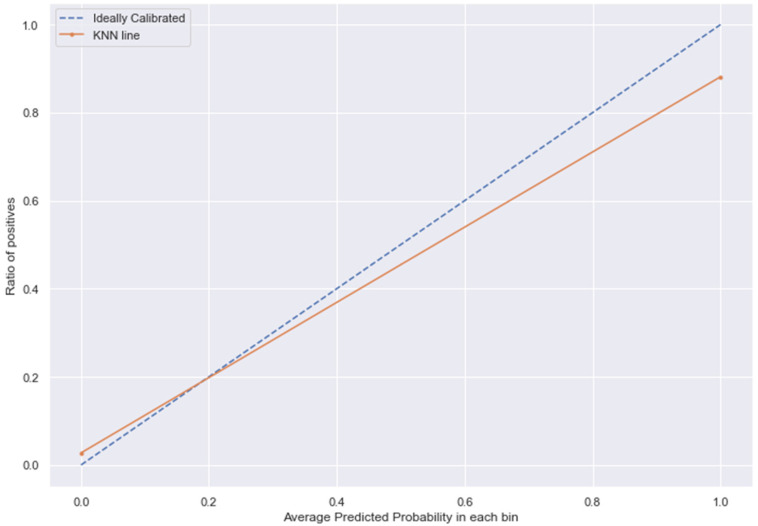
KNN line without calibration vs. after calibration (Ideal fit is represented as blue line). Note: All other models (RFC, GBC, BC) were overfitting the data and had a balanced accuracy score of 1.

**Figure 17 medicina-58-01568-f017:**
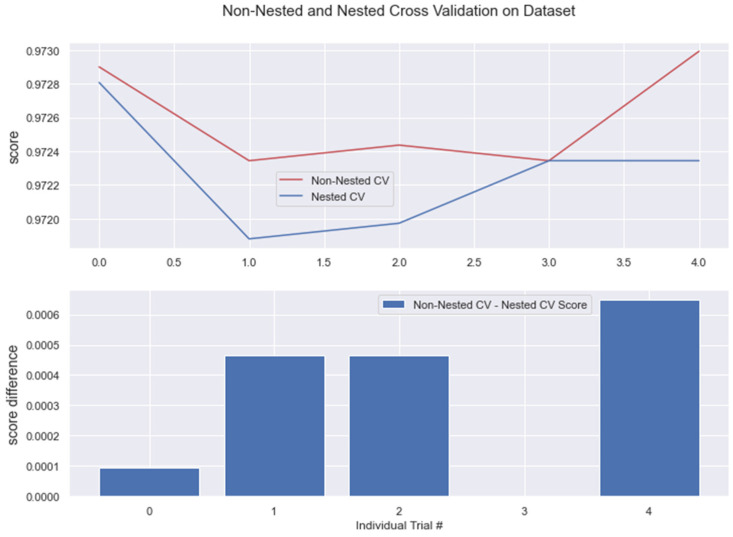
Non-Nested and Nested cross-validation on LR algorithm.

**Figure 18 medicina-58-01568-f018:**
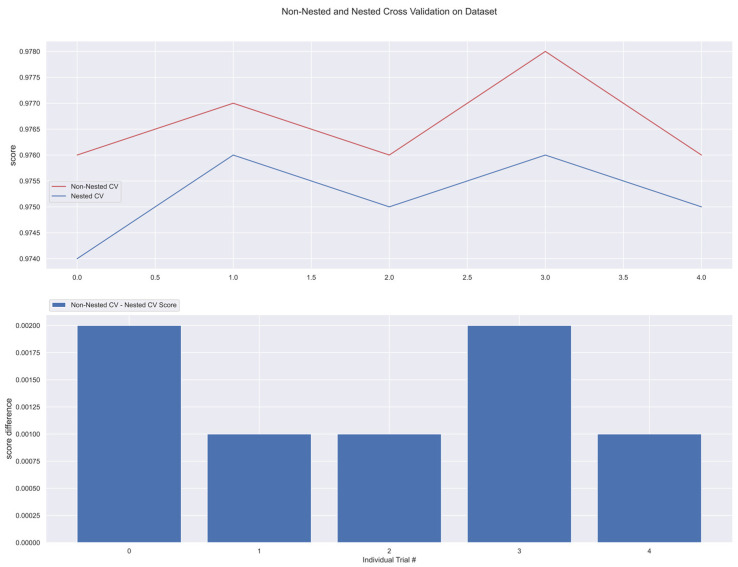
Non-Nested and Nested cross-validation on SVM algorithm.

**Figure 19 medicina-58-01568-f019:**
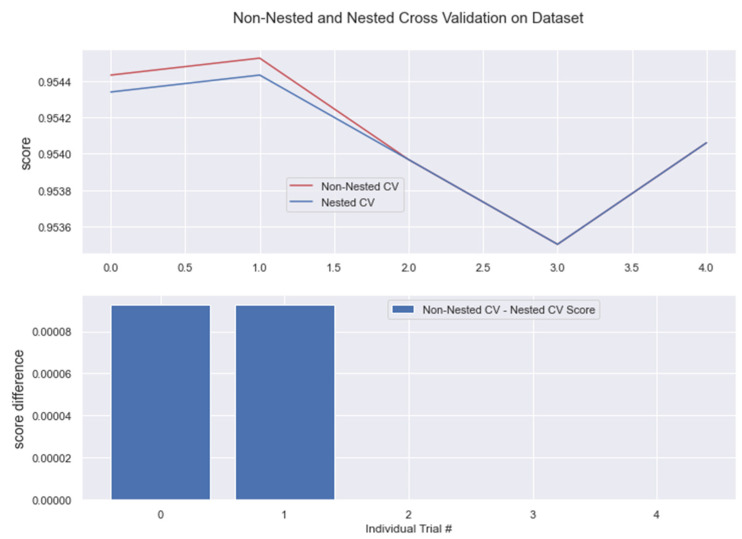
Non-Nested and Nested cross-validation on RC algorithm.

**Figure 20 medicina-58-01568-f020:**
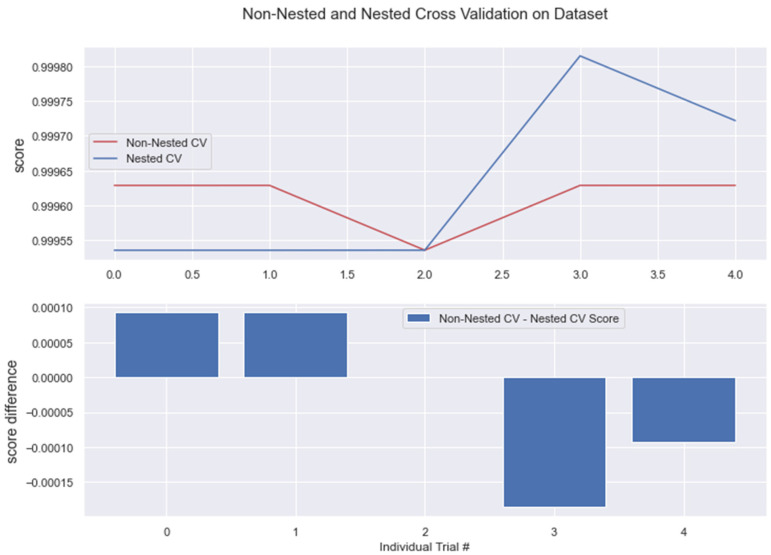
Non-Nested and Nested cross-validation on RFC algorithm. Note: All other algorithms (KNN, BC, GBC) have the same non-nested and nested cv scores. This is represented in the below figure.

**Figure 21 medicina-58-01568-f021:**
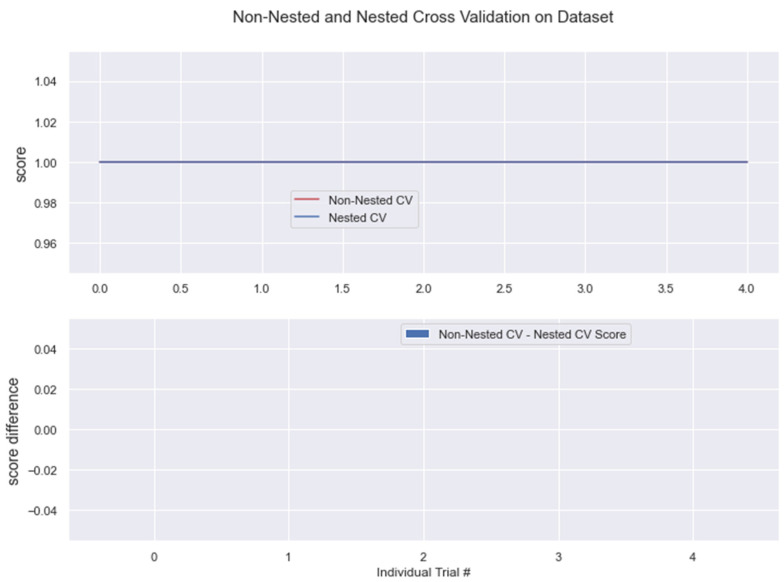
Algorithms having the same Non-nested and Nested cv score.

**Table 1 medicina-58-01568-t001:** Demographic characteristics of the patients.

Characteristics	Patients with T2DM *n* = 25,868	Patients with HTN *n* = 32,750	Patients Having Both *n* = 6419	Total Patients (T2DM + HTN + Both) *n* = 65,037	Total Patients Used for Prediction Model (T2DM or HTN) *N* = 58,618	Statistical Significance
**Sex**						0.136
Male	13,138 (50.79)	16,849 (51.45)	3070 (47.83)	33,057 (50.83)	29,987 (51.16)	
Female	12,730 (49.21)	15,901 (48.55)	3349 (52.17)	31,980 (49.17)	28,631 (48.84)	
**Age (years) (mean ± SD)**	75.05 ± 13.41	75.19 ± 13.84	75.49 ± 11.89	75.16 ± 13.49	75.12 ± 13.65	0.808
<35	94 (0.36)	150 (0.46)	8 (0.12)	252 (0.39)	244 (0.42)	
35–57	2408 (9.31)	3236 (9.89)	417 (6.50)	6061 (9.32)	5644 (9.63)	
58–80	13,307 (51.44)	16,157 (49.33)	3564 (55.52)	33,028 (50.78)	29,464 (50.26)	
>80	10.059 (38.89)	13,207 (40.33)	2430 (37.86)	25,696 (39.51)	13,217 (22.55)	
**Discharge status**						0.001 *
Treatment and discharge	1142 (4.41)	1953 (5.96)	325 (5.06)	3420 (5.26)	3095 (5.28)	
Continue to be hospitalized	0	0	0	0	0	
Change to outpatient treatment	22,798 (88.13)	29,422 (89.84)	5859 (91.28)	58,079 (89.30)	22,827 (38.94)	
Death	585 (2.26)	298 (0.91)	30 (0.47)	913 (1.40)	883 (1.51)	
Automatic discharge	648 (2.51)	584 (1.78)	125 (1.95)	1357 (2.09)	1232 (2.10)	
Transfer	318 (1.23)	268 (0.82)	56 (0.87)	642 (0.99)	586 (0.999)	
Change of identity	0	0	0	0	0	
Absconding	3 (0.01)	2 (0.006)	0	5 (0.01)	5 (0.009)	
Suicide	1 (0.003)	0	0	1 (0.001)	1 (0.002)	
Other	373 (1.44)	223 (0.68)	24 (0.37)	620 (0.95)	596 (1.02)	
**No. of comorbidities**						-
0	123 (0.48)	258 (0.79)	0 (0)	381 (0.59)	381 (0.65)	
1	2425 (9.37)	6858 (20.94)	0 (0)	9283 (14.27)	9283 (15.84)	
2	11,269 (43.56)	15,757 (48.11)	134 (2.09)	27,160 (41.76)	27,026 (46.11)	
≥3	12,051 (46.59)	9877 (30.16)	6285 (97.91)	28,213 (43.38)	21,928 (37.41)	
**Hospital Cost**						0.141
Average Cost(min–max)	13,208 (0–1,212,764)	10,449 (0–768,724)	10,120 (0–768,724)	11,514 (0–1,212,764)	11666 (0–1,212,764)	
Median (IQR)	7962 (4470–13,397)	6852 (3947–11,107)	6895 (4119–11,084)	7228 (4298–11,972)	7228 (4298–12,185)	
**LoS**						0.031 *
Average LoS (min–max)	8.46 (0–6059)	6.56 (0–3087)	6.60 (0–1887)	7.32 (0–6059)	6.60 (0–1887)	
Median (IQR)	5.00 (3.00–8.00)	4.00 (2.00–7.00)	4.00 (3.00–7.00)	5.00 (3.00–8.00)	4.00 (3.00–7.00)	

Note—Length of stay (LoS), interquartile range (IQR), maximum (max), minimum (min). Continuous values were recorded as median (1st–3rd quantile), and categorical values were recorded as absolute numbers and percentages; * The difference is significant for *p* value < 0.05.

**Table 2 medicina-58-01568-t002:** Most common comorbidities with T2DM and HTN.

Common Comorbidities
Metabolic disorders Coronary artery disease Myocardial infarction Stroke Congestive heart failure Aneurysm Pneumonia Urinary tract infection Inguinal hernia Intracerebral hemorrhage

**Table 3 medicina-58-01568-t003:** LoS prediction performance of various models.

Model	MSE	RMSE	MAE	R2
SVM	0.393	0.510	0.121	0.486
LR	0.570	0.755	0.065	0.172
GBM	0.584	0.755	0.004	0.397
XGBoost	0.312	0.386	0.123	0.633
RF	0.261	0.401	0.027	0.591

**Table 4 medicina-58-01568-t004:** Mortality prediction performance of various models for classification model.

Classifier	Accuracy Score	Balanced Accuracy Score	Test Score	Precision	Recall	AUC	AUPR
LoR	0.9779	0.9719	0.9728	0.9432	0.9786	0.97	0.93
RC	0.9736	0.9592	0.9692	0.9312	0.9463	0.94	0.89
SVM	0.7899	0.7562	0.7332	0.7599	0.6524	0.88	0.89

Note: Models which were not included in the above tables were either overfit models or less accurate in predicting LoS and mortality. This can be further understood in Table 5.

**Table 5 medicina-58-01568-t005:** Excluded models based on overfitting or accuracy.

Model Name	Precision	Recall	Train Accuracy	Test Accuracy	F1 Score	AUC
Decision Tree	1.00	1.00	1.00	1.00	1.00	1.000000
Random Forest	1.00	1.00	1.00	1.00	1.00	0.999827
Logistic Regression	0.94	0.98	0.97	0.97	0.96	0.971884
Ada Boost	1.00	1.00	1.00	1.00	1.00	1.000000
Bagging	1.00	1.00	1.00	1.00	1.00	1.000000
Gradient Boosting	1.00	1.00	1.00	1.00	1.00	1.000000
XGB	1.00	1.00	1.00	1.00	1.00	1.000000
SVC	0.82	0.49	0.77	0.77	0.61	0.710974
K-Neighbors	0.85	0.95	0.95	0.92	0.90	0.926795
Gaussian	1.00	0.98	0.99	0.99	0.99	0.992494

## Data Availability

Although anonymized data were used, the data that support the findings of this study are not publicly available. However, aggregated data are available from authors upon reasonable request and with the permission of NHIRD.

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
