# Peer review of "A Machine Learning Model to Predict Length of Stay and Mortality among Diabetes and Hypertension Inpatients"

_medicina, 2022, doi:10.3390/medicina58111568_

Round 1

Reviewer 1 Report (Previous Reviewer 2)

The work has been significantly improved with respect to the previous version. However, I still have some concerns regarding the presentation of the results. In particular, I suggest the authors enhance the representation of Figures 17 and 18.

In addition:

- in the Discussion (line 862), the term learning should be added: "using several machine learning techniques"

- in the Discussion (from line 936), the authors introduce the correlation of T2DM with increased risk of cardiovascular and microvascular diseases. Several papers from Maulucci et al analyze the possibility of evaluating this risk in terms of membrane fluidity. I think that the authors should include them in their observations.

Author Response

Point 1 The work has been significantly improved with respect to the previous version. However, I still have some concerns regarding the presentation of the results. In particular, I suggest the authors enhance the representation of Figures 17 and 18.

Response 1: Thank you for pointing this out. We have now enhanced the figures 17 and 18 in the manuscript.

Point 2 In the Discussion (line 862), the term learning should be added: "using several machine learning techniques"

Response 2: Thank you for your suggestion. We have added the term learning in the line.

Point 3 in the Discussion (from line 936), the authors introduce the correlation of T2DM with increased risk of cardiovascular and microvascular diseases. Several papers from Maulucci et al analyze the possibility of evaluating this risk in terms of membrane fluidity. I think that the authors should include them in their observations.

Response 3: Thank you for your valuable suggestion. We have added the required references :[Ref 37] and [Ref 38] to the line.

[37] Maulucci, G.; Cordelli, E.; Rizzi, A.; De Leva, F.; Papi, M.; Ciasca, G.; Samengo, D.; Pani, G.; Pitocco, D.; Soda, P. Phase separation of the plasma membrane in human red blood cells as a potential tool for diagnosis and progression monitoring of type 1 diabetes mellitus. PLoS One 2017, 12, e0184109

[38]à Cordelli, E.; Maulucci, G.; De Spirito, M.; Rizzi, A.; Pitocco, D.; Soda, P. A decision support system for type 1 diabetes mellitus diagnostics based on dual channel analysis of red blood cell membrane fluidity. Computer methods and programs in biomedicine 2018, 162, 263-271.

Reviewer 2 Report (Previous Reviewer 1)

The technical quality of the manuscript has been improved. The use of nested cross-validation is to be appreciated. The software used is now reported, but it is difficult to understand, because both Python and R are used, with many libraries, and the purpose of some of them is not obvious. For instance, why did the authors used both mlr and mlr3 (as mlr3 is just an improved version of mlr)? It is also not clear why they used both mlr and caret (unless clarified for which pre-processing or post-processing or for what algorithms each of them were applied). A bit of clarification in this direction would be necessary.

I tend to be skeptical about the optimal results of only univariate and bivariate analysis for feature selection. But at least is clearer than in the previous version of the manuscript.

It is a pity that data are not made available by the authors. In this context it is difficult for someone to check the accuracy of the model. I find it concerning that in nested-cross validation ROC of 1 and balanced accuracy values very close to 1 are finding, which make me doubt the validity of the models. The authors, however, should at least release the computer code used to analyze the data, so as to allow any informed reader to have an idea on the correctness of the code use.

Author Response

Point 1 The technical quality of the manuscript has been improved. The use of nested cross-validation is to be appreciated. The software used is now reported, but it is difficult to understand, because both Python and R are used, with many libraries, and the purpose of some of them is not obvious. For instance, why did the authors used both mlr and mlr3 (as mlr3 is just an improved version of mlr)? It is also not clear why they used both mlr and caret (unless clarified for which pre-processing or post-processing or for what algorithms each of them were applied). A bit of clarification in this direction would be necessary.

Response 1: Thank you for bringing this to our attention. We have removed mlr3 and caret from the software and libraries section because they were not used for analysis. We imported these libraries but never used them for analysis.

Point 2 I tend to be skeptical about the optimal results of only univariate and bivariate analysis for feature selection. But at least is clearer than in the previous version of the manuscript.

Response 2: Thank you for your comment.

Point 3 It is a pity that data are not made available by the authors. In this context it is difficult for someone to check the accuracy of the model. I find it concerning that in nested-cross validation ROC of 1 and balanced accuracy values very close to 1 are finding, which make me doubt the validity of the models. The authors, however, should at least release the computer code used to analyze the data, so as to allow any informed reader to have an idea on the correctness of the code use.

Response 3: Thank you for providing your comments. Algorithms such as RC, KNN, BC, and GBC have a CV score of 1 (Overfit). Other models have a value close to 1. We can use SVM with calibration as an alternative because we achieved a balanced accuracy of 0.85.

This manuscript is a resubmission of an earlier submission. The following is a list of the peer review reports and author responses from that submission.

Round 1

Reviewer 1 Report

The LOS abbreviation in the abstract should be explained with first use (length of stay?)

“9,996 people died in Taiwan from  DM, with 2,736 instances reported in people 85 years of age and older”. This sentence can only be interpreted and be meaningful if the period is specified (in a month? In a year? In what period?)

“In general, 70%  of patients with T2DM had HTN, and patients with previous HTN were 2.5 times more  likely to develop T2DM”. It should be better clarified that this statement is found in the literature, because the data from the previous sentence are not in agreement with this finding.

“The  overall  missingness  in  each  of  the  features  was  less  than  10%,  so  we  ultimately removed all the missing values from our study.” However, the flow chart does not include this aspect, and because the missing data do not necessarily are the same for all patients, the number of subjects with missing data could be higher than 10% of all sample. Moreover, even 10% of 65000 would mean 6500 patients that should not be present in the data analysis. However, the authors state that they have analyzed data from 65,037 patients. These aspects should be clarified.

Predictive model development and evaluation: The authors only used simple 10-fold cross-validation, which is not sufficient to control bias. A nested (double) cross-validation should be used (see e.g. this widely known resource https://scikit-learn.org/stable/auto_examples/model_selection/plot_nested_cross_validation_iris.html)

“The Receiver Operating Characteristics (ROC)  Area Under the Curve (AUC), F-beta, Precision, Recall, Cross Validation score, Accuracy  score,  Test  score  and  Area  Under  Precision-Recall  (AUPR)  were  used  to  evaluate  the  models.” Apparently these are only metrics for classification, but none are mentioned for regression. It is also imperative that the authors generate and present calibration plots to allow the assessment of the classification models. Section 3.8 shows a calibration plot only for the SVM model. “Accuracy score” is also highly influenced by the data imbalance, therefore the authors should use balanced accuracy instead of simple accuracy.

The authors state that they had 67 features available. Were all 67 used? Was there any feature selection method used to reduce the number of features?

The software and libraries used to run the models should be clarified.

Unlike MSE, RMSE, and MAE, R2 (coefficient of determination) needs to be maximized for better performance. Therefore, an R2 of 0.172 is pretty useless. Even the highest R2 value (0.302) is much too low to claim practical use.

“Table 4 indicates the mortality prediction performance.” Results from (nested) cross-validation should also be provided and it should be clarified whether Table 4 is based on the training data set or on the test set.

Figures 4 and 5: it is not clear to which of the model(s) they refer to.

“Our current findings indicated that the inpatient cost of patients with T2DM exceeds  patients with HTN.” Nothing in the results seem to support this. The authors should therefore cross-reference to the results section to allow the reader to understand the data support for this statement.

Author Response

We are extremely thankful to the reviewer for his/her valuable comments and suggestions that have inspired several changes in our work and we hope to have significantly improved it.  Please find below, the point-to-point response to the reviewer’s suggestions.

1 - The LOS abbreviation in the abstract should be explained with first use (length of stay?)

Response: LOS abbreviation has been explained in the abstract.

2 - “9,996 people died in Taiwan from  DM, with 2,736 instances reported in people 85 years of age and older”. This sentence can only be interpreted and be meaningful if the period is specified (in a month? In a year? In what period?)

Response: The required time details have been added in the sentence. 

3 - “In general, 70%  of patients with T2DM had HTN, and patients with previous HTN were 2.5 times more  likely to develop T2DM”. It should be better clarified that this statement is found in the literature, because the data from the previous sentence are not in agreement with this finding.

Response: The data was referred from below 2 research papers - 

https://www.ncbi.nlm.nih.gov/pmc/articles/PMC3785394/

https://www.ncbi.nlm.nih.gov/pmc/articles/PMC5305248/

The sentence has been further improved to 

In general, HTN is prevalent among 70% of T2DM patients, whereas patients with HTN are 2.5 times more likely to develop T2DM as a primary comorbidity

4 - “The  overall  missingness  in  each  of  the  features  was  less  than  10%,  so  we  ultimately removed all the missing values from our study.” However, the flow chart does not include this aspect, and because the missing data do not necessarily are the same for all patients, the number of subjects with missing data could be higher than 10% of all sample. Moreover, even 10% of 65000 would mean 6500 patients that should not be present in the data analysis. However, the authors state that they have analyzed data from 65,037 patients. These aspects should be clarified.

Response: The table explains the criteria used for selecting the patients. The data does not contain patients with both T2DM and HTN.

5 - Predictive model development and evaluation: The authors only used simple 10-fold cross-validation, which is not sufficient to control bias. A nested (double) cross-validation should be used (see e.g. this widely known resource https://scikit-learn.org/stable/auto_examples/model_selection/plot_nested_cross_validation_iris.html)

Response: The double cross validation has been implemented now.

6 - “The Receiver Operating Characteristics (ROC)  Area Under the Curve (AUC), F-beta, Precision, Recall, Cross Validation score, Accuracy  score,  Test  score  and  Area  Under  Precision-Recall  (AUPR)  were  used  to  evaluate  the  models.” Apparently these are only metrics for classification, but none are mentioned for regression. It is also imperative that the authors generate and present calibration plots to allow the assessment of the classification models. Section 3.8 shows a calibration plot only for the SVM model. “Accuracy score” is also highly influenced by the data imbalance, therefore the authors should use balanced accuracy instead of simple accuracy.

Response: The graphs for the regression model are done by co-authors. Calibration is done only on SVM to check the difference in accuracy. We found that this approach results in decrease in accuracy, precision and recall of the model. Hence, we did not do it for other models. Instead we plotted precision-recall and AUC-ROC. The balanced accuracy in all the models has been implemented now..

7 - The authors state that they had 67 features available. Were all 67 used? Was there any feature selection method used to reduce the number of features?

Response: We didn't use all 67 features. 27 features were taken into consideration for mortality prediction. Top 15 features  of mortality were ranked by using Random Forest ClassifierFor LOS, 24 features were selected. Top 15 features of LoS were ranked by using the chi-square method. 

8 - The software and libraries used to run the models should be clarified.

Response: A section to document softwares and libraries used has been created.

9 - Unlike MSE, RMSE, and MAE, R2 (coefficient of determination) needs to be maximized for better performance. Therefore, an R2 of 0.172 is pretty useless. Even the highest R2 value (0.302) is much too low to claim practical use.

Response: MSE, RMSE, and MAE, R2 is done in LOS prediction.

10 - “Table 4 indicates the mortality prediction performance.” Results from (nested) cross-validation should also be provided and it should be clarified whether Table 4 is based on the training data set or on the test set.

Response: The table from both CV and nestedCV has been provided and the training / validation set has also been mentioned. 

11 - Figures 4 and 5: it is not clear to which of the model(s) they refer to.

Response: Figure 4 and 5 have been clarified for related model(s). 

12 - “Our current findings indicated that the inpatient cost of patients with T2DM exceeds  patients with HTN.” Nothing in the results seem to support this. The authors should therefore cross-reference to the results section to allow the reader to understand the data support for this statement.

Response: This was a part of preliminary analysis. Now, we have done the calculation by selecting a cohort and counted the number of days in hospital which escalate cost. 

Reviewer 2 Report

In this paper, Barsasella et al. developed a machine learning-based model to predict the length of stay and mortality among hospitalized patients with type 2 diabetes and/or hypertension in Taiwan. Although the work is of potential scientific interest and impact, helping in health care management and intervention and resource planning.

Here are my comments:

- The introduction is not clear. Lots of information, numbers, and data are reported, but it lacks a proper structure. In this form, the reader cannot understand the main objective and aim of the presented study.

- The results are bad organized. In Table 1, a legend should be included to explain the meaning of the numbers in brackets. Moreover, a statistical comparison among groups is missing.

- All the figures lack the caption. Results are not explained, and graphs are shown without being described. Figure 2 should be changed into a table. In Figure 3 axes labels are missing. What does the x-axis represent? Moreover, all the figures should be presented in the same style. Graphs in Figure 6 and Figure 8 to 13 are not readable. Figure 12 is missing.

- In the discussion, the novelty and scientific significance of the obtained and presented results are not properly described.

I think that the work should be totally restructured and the results should be better organized, presented, and discussed.

Author Response

Reviewer2

We are extremely thankful to the reviewer for his/her valuable comments and suggestions that have inspired several changes in our work and we hope to have significantly improved it.  Please find below, the point-to-point response to the reviewer’s suggestions.

In this paper, Barsasella et al. developed a machine learning-based model to predict the length of stay and mortality among hospitalized patients with type 2 diabetes and/or hypertension in Taiwan. Although the work is of potential scientific interest and impact, helping in health care management and intervention and resource planning.

Response: Thank you for providing your valuable feedback. All of the mentioned comments have been addressed. Please see the response for each comment one by one.

Here are my comments:

1 - The introduction is not clear. Lots of information, numbers, and data are reported, but it lacks a proper structure. In this form, the reader cannot understand the main objective and aim of the presented study.

Response: The Introduction section has been revised completely. Relevant information has been added and the structure is organized. The motivations and objectives of the study have been clearly defined.

2 - The results are bad organized. In Table 1, a legend should be included to explain the meaning of the numbers in brackets. Moreover, a statistical comparison among groups is missing.

Response: The Results section has been organized with more details and interpretation. Legend for Table 1 has been added to explain the meaning of numbers. Statistical comparison among groups has been added.

3 - All the figures lack the caption. Results are not explained, and graphs are shown without being described. Figure 2 should be changed into a table. In Figure 3 axes labels are missing. What does the x-axis represent? Moreover, all the figures should be presented in the same style. Graphs in Figure 6 and Figure 8 to 13 are not readable. Figure 12 is missing.

Response: All the figures have been provided with captions. Results are explained in more details. Figure 2 has been changed to Table 5. Axes labels of Figure 3 have been added, and axes details have been provided. All figures have been presented with consistent style. Graphs for Figure 6 and 8 till 13 have been revised. All figure numbers are identified clearly, so there is no missing figure number now.

4 - In the discussion, the novelty and scientific significance of the obtained and presented results are not properly described.

Response: The results have been interpreted and their novelty and significance has been discussed by doing a comparative analysis with the related literature.

5 - I think that the work should be totally restructured and the results should be better organized, presented, and discussed.

Response: The whole paper has been structured and improved significantly. The results have been organized, interpreted, and presented clearly for better understanding.

Reviewer 3 Report

Good work. No comment

Author Response

We are extremely thankful to the reviewer for his/her valuable comments.